# Structural Origins of Poor Health Outcomes in Documented Temporary Foreign Workers and Refugees in High-Income Countries: A Review

**DOI:** 10.3390/healthcare11091295

**Published:** 2023-05-01

**Authors:** Borum Yang, Clara Kelly, Isdore Chola Shamputa, Kimberley Barker, Duyen Thi Kim Nguyen

**Affiliations:** 1Faculty of Medicine, Dalhousie University NB, Saint John, NB E2L 4L5, Canada; 2Department of Nursing & Health Sciences, University of NB, Saint John, NB E2L 4L5, Canada; 3Department of Public Health, Government of NB, Saint John, NB E1A E9H, Canada; 4Faculty of Business, University of NB, Saint John, NB E2L 4L5, Canada

**Keywords:** structural discrimination, structural factors, health disparities, healthcare access, refugees, documented temporary foreign workers

## Abstract

Despite growing evidence of racial and institutional discrimination on minoritized communities and its negative effect on health, there are still gaps in the current literature identifying health disparities among minoritized communities. This review aims to identify health barriers faced by relatively less studied migrant subgroups including documented temporary foreign workers and refugees residing in high-income Organisation for Economic Co-operation and Development (OECD) countries focusing on the structural origins of differential health outcomes. We searched Medline, CINAHL, and Embase databases for papers describing health barriers for these groups published in English between 1 January 2011 and 30 July 2021. Two independent reviewers conducted a title, abstract, and full text screening with any discrepancies resolved by consensus or a third reviewer. Extracted data were analyzed using an inductive thematic analysis. Of the 381 articles that underwent full-text review, 27 articles were included in this review. We identified housing conditions, immigration policies, structural discrimination, and exploitative labour practices as the four major emerging themes that impacted the health and the access to healthcare services of our study populations. Our findings highlight the multidimensional nature of health inequities among migrant populations and a need to examine how the broader context of these factors influence their daily experiences.

## 1. Introduction

The proportion of international migrants has risen over the past five decades worldwide, with an estimated 272 million people living in a country other than their country of birth [1]. Notably, countries such as Canada have adopted immigration as a key means to support population, economic, and cultural growth [1,2]. For instance, Canada issued over 98,310 documented temporary foreign worker permits in 2019, and resettled 30,087 refugees, the highest number of any country worldwide [2].

Evidence of the differential health outcomes of migrants compared to the native-born population have long been well-established in both the United States and Canada [3,4].

In fact, the phenomena of racial and institutional discrimination on minoritized communities and its negative effects on health has been documented as early as the 1990s when the structural origins of poor health outcomes (e.g., poor regulation protections from pesticides) were identified [5]. However, despite growing evidence, public discourse and the medical literature community has been slow to identify structural racism as a root cause of health disparities [6,7]. This is certainly true regarding many reviews and research published to date. For example, Parajuli and Horey [8], Lane et al. [9], and Cheng et al. [10] revealed that more often than not, major themes regarding barriers to healthcare access pointed to individual characteristics and cultural factors (e.g., nutrition deficits, families’ conceptualization of life in Canada, and health literacy) as the primary drivers of disparities. Even the identified systemic barriers are limited to factors such as a lack of care-provider cultural competency, service time constraints, and policy issues regarding funding limits and the inadequate design of the healthcare system. Moreover, there are acknowledged gaps for subgroups of migrants; specifically, Chowdhury et al. [11] and Salami et al. [12] reported a dearth of literature on the health of migrants with disabilities and of documented temporary workers outside the agricultural sector.

The overarching goal of this study was to identify health barriers faced by relatively less-studied migrant subgroups including documented temporary foreign workers and refugees with a particular focus on the structural origins of differential health outcomes. These data will be used to inform the development a proposed newcomer health clinic in Southern New Brunswick, Canada, aimed to provide improved healthcare access to newcomers. Due to the limited literature available within Canada, we broadened the scope to documented migrants residing in high-income Organisation for Economic Co-operation and Development (OECD) countries. We also focused on research published in the past decade to provide an update on the literature. For the purposes of this review, the term structural factors is used to describe the term structural racism as conceptualized by Krieger [13] (p. 645): “the totality of ways in which societies foster discrimination, via mutually reinforcing [inequitable] systems… (e.g., in housing, education, employment, earnings, benefits, credit, media, healthcare, criminal justice, etc.) that in turn reinforce discriminatory beliefs, values, and distribution of resources”.

## 2. Materials and Methods

### 2.1. Study Design

Thus, to provide timely access to evidence-based health information, we used a review methodology guided by the Cochrane Handbook for systematic Reviews of Interventions [14]. This included defining a research question, conducting a search based on predetermined eligibility criteria and search strategy, selecting articles, extracting data, and a knowledge synthesis. A protocol was not registered for this review. A preliminary review of the literature was conducted to identify a list of key terms used to describe our sample and phenomenon of interest. (See Appendix A for the full list of key terms in the search string). Using the key terms identified, we conducted a comprehensive search to help inform the scope of pre-existing literature and find gaps in knowledge. The PICoS framework (Table 1), an alternative version of PICO modified for qualitative studies, was used to design the research question and the inclusion and exclusion criteria [15].

### 2.2. Search Strategy

The search strategy included a combination of subject headings and synonyms for each category under the PICoS framework (see Appendix A for the full search strategy). The criterion for selecting the search terms was to include the preselected articles found in our preliminary search. An extensive list of terms was used to ensure the search was comprehensive. We searched three electronic databases, Medline, Cumulative Index of Nursing & Allied Health Literature (CINAHL), and Embase, based on their ready access to evidence-based content on the research topic, and robust search capability with controlled vocabulary and keywords. Searches were conducted on 19 July 2021 with a date limitation of 1 January 2011 to July 2021.

### 2.3. Inclusion and Exclusion Criteria

Qualitative and quantitative peer-reviewed primary studies published from 1 January 2011 to July 2021 were chosen to give priority to the more recent knowledge synthesis and to ensure a wide range of papers could be included, due to the lack of research in this area. Other inclusion criteria included research with (1) peer-reviewed articles written in English, (2) conducted in a high-income OECD country, and (3) whose study population of interest was documented temporary foreign workers and/or refugees. We chose to include studies looking at both the perspectives of the population in question or actors working in the field of migrant health (see Appendix B for actors working in the field of migrant health). We also chose to only include papers whose topic of interest was primarily health barriers and excluded those that did not have a specific focus on health barriers. We focused on articles written in English due to the language capabilities of the authors. In addition, we only included articles from high-income OECD countries to allow a better comparison to the country in which the results of this paper will be used (Canada). Papers that were not primary research, such as reviews, presentations, and editorial articles, were excluded to ensure rigor. Additionally, papers were excluded if the sample population was listed as a general migrant population or did not specify if it included other groups such as economic immigrants, nondocumented migrants, or groups not included in our population of interest. The full list of inclusion and exclusion criteria is presented in Table 2.

### 2.4. Data Selection

A search of the electronic databases identified a total of 5905 articles. Figure 1 outlines the selection process of articles as per the PRISMA guidelines. All search results were imported into Covidence. After removing duplicates (n = 1940), 3969 articles were identified. A pilot exercise using the same 10 abstracts and full-text articles was conducted to calibrate and ensure a standardization between author 1 and author 2. Then, author 1 and author 2 independently performed both title and abstract screening and full-text review; 277 conflicts were identified, which were resolved either by discussion or a third reviewer; the inter-rater reliability rate was 90.4% (3588/3969). Of the 381 articles that underwent full-text review, 27 articles were included in this review.

### 2.5. Data Extraction and Analysis

We developed an extraction tool adopted from the Cochrane handbook [14]. Extracted data included authors, publication year, study design, study location, research question, participant characteristics, size, and main findings (Appendix B). To systematically identify and synthesize findings across all included studies, we used a thematic analysis [16]. We used NVivo 12 Pro software to carry out the coding process. The first phase of analysis included familiarization, during which each paper was read multiple times, and any potential data of interest regarding health barriers were highlighted. For qualitative studies, the data of interest included interview quotations and paraphrases regarding barriers or needs our study population experienced in accessing healthcare. We included the perspectives of actors working in the field of migrant health to provide additional perspectives. Actors working in the field of migrant health included physicians, nurses, allied healthcare professionals, policymakers, advocacy agency workers, and others. The full list of the actors working in the field of migrant health can be found in Appendix B. For quantitative studies, the data of interest included variables significantly associated with the health status of the sample populations. Once this was done for all papers, the identified extracts were analyzed and labeled with a “code” that provided a definition or an interpretation. This process was repeated until all the papers were fully coded. The second phase comprised exploring different ways to combine the codes by drawing thematic maps and searching for unifying themes and patterns across the entire data set. Priority was given to themes addressing the structural origins of poor health outcomes. This was done for each population separately and additionally across both populations to identify the existence of similarities or differences among the two groups.

## 3. Results

### 3.1. Study Characteristics

Of the 27 papers included, 18 focused on refugee challenges, and 9 on documented temporary foreign worker challenges (see Table 3). Most studies were published from 2017 onwards (n = 22). Studies were mostly descriptive in nature, using a qualitative (n = 21), quantitative (n = 1), and mixed-methods (n = 5) approach. Most studies were conducted in Canada (n = 13), followed by the United States (n = 6), Australia (n = 3), New Zealand (n = 2), and one each from the Netherlands and Germany. Most studies discussed multiple ethnicities and presented them from various perspectives. Below, we describe in further detail the main barriers each group faced. Excerpts from selected papers demonstrating the emerging themes reported herein are presented in Table 4.

### 3.2. Barriers for Documented Temporary Foreign Workers

We identified nine studies in total that looked at documented temporary foreign workers; four studies examined caregivers, two examined agricultural workers, and one on various industries including meat processing, food services, hospitality, and construction. Two studies interviewed the perspectives of key informers. We discuss the four identified barriers below: housing conditions, immigration policies, structural discrimination, and exploitative labour practices.

#### 3.2.1. Housing Conditions

Our results showed that documented temporary foreign workers across different fields lived in suboptimal and unsafe conditions, including overcrowded housing, basements, and rooms without beds or windows [17,19,20,26]. Despite the removal of the “live-in” requirement in 2014 by the Canadian government, most caregivers still lived with their employers either because of the preference of the employers or the workers could not afford the high cost of living alone [27]. Caregivers noted significant distress related to the lack of privacy from an inability to lock doors to their bedrooms, being monitored by security cameras, and being hassled by the children of their host families [17,28] There was significant psychological stress noted because of the lack of socialization due to long work hours, remoteness of the homes, and limited transportation [18,26].

#### 3.2.2. Immigration Policies

A very common barrier identified for documented temporary foreign workers was regulations and government policies that placed them in precarious positions that directly impaired their access to care. Advocates and healthcare workers reported documented temporary foreign workers hiding their illness or choosing not to seek medical care to maintain a clean medical record necessary for a successful immigration application [17,29]. We found that many workers were aware of the exploitative working conditions but chose to accept them as they viewed obtaining permanent residency as a way towards a better future for their families [18].

The vulnerability of migrant workers to exploitative conditions was largely enabled by the absence of sufficient oversight and enforcement by regulatory bodies. Seasonal agricultural workers reported that workplace inspections were announced to the employers beforehand, which allowed hazardous and poor working conditions to continue unchecked [18,20]. Caregivers expressed that the lack of regulations exacerbated the uneven power dynamic between employer and caregiver, leaving them no choice but to tolerate hazardous work [18]. A number of immigrant service providers as well as policymakers noted that while policymakers expect the employers to take care of health needs of their workers, there is no policy mandating them to provide such services [17]. This was apparent in the numbers: despite a federal government requirement that employers are responsible for providing third-party health insurance for employees during their first three months, only 24% of caregivers in a study were found to have one [27].

In our review, only one group of documented temporary foreign workers reported significantly different experiences. Workers in the meat processing industry reported that the union provided them with an opportunity to advocate for their needs and have them resolved [20]. The union representative stressed the need for unions among documented temporary foreign workers because their precarious position left them vulnerable to exploitation [20].

#### 3.2.3. Structural Discrimination

Our final theme regarding barriers for documented temporary foreign workers includes structural discrimination. Workers reported having limited access to social and public spheres of life due to various forms of cultural and systemic exclusions. They reported experiencing intolerance from employers and struggled with expressed feelings of “cold indifference” despite efforts to be recognized as members of the community [28] (p. 212). Workers reported experiencing racial profiling during contact with law enforcement [23]. Caregivers reported experiencing isolation and alienation due to deskilling and downward social mobility experienced in their host countries, evidenced by facing verbal abuse from employers who harbored negative stereotypes [17,28,29].

#### 3.2.4. Exploitative Labour Practices

The most common barrier for documented temporary foreign workers examined was exploitative labour practices. We found that working conditions across all groups of workers were hazardous and unsafe, often involving contract breaches related to working hours and job descriptions. Workers were often asked to work through breaks, work without proper equipment, work overtime hours without pay, and perform tasks beyond their responsibilities and skills [17,19,20,26,27,28]. Notably, seasonal agricultural workers and caregivers reported long work hours as a major barrier to accessing health services when needed [17,20,26]. Employers were often found to use threats of job termination and deportation as leverage to maintain exploitative labor conditions [17,18,19,20,29,30]. Agricultural workers and healthcare workers reported being persuaded against or actively prevented from seeking medical care or accessing worker’s compensation by employers worried about incurring costs, with some even being harassed for reporting an injury [18,19,20,30].

### 3.3. Barriers for Refugees

We identified 18 studies regarding refugees: 5 studies looked at the perspectives of refugees; 8 studies considered the perspectives of actors working in the field of migrant health including but not limited to physicians, nurses, allied healthcare professionals, policymakers, and advocacy agency workers; and 5 studies examined both the perspectives of refugees and the actors working in the field of migrant health. The three main themes were: housing conditions, immigration policies, and structural discrimination.

#### 3.3.1. Housing Conditions

Another common health barrier we found was poor housing conditions. Results showed that the social context of migrants negatively impacted the refugees’ health by limiting access to social and economic opportunities. Refugees expressed a fear of hearing and witnessing gunshots, drug activities in the neighbourhood, and animals in their houses, which not only distracted them from tending to their health but also prevented them from venturing outside their homes [21]. In instances where refugees were located far from services, many reported ignoring their illness because they were unable to navigate the city transportation [25,31,32] Information from the literature indicated that refugees with disabilities were rarely provided with adequate accommodations related to their needs such as housing with elevators for those in [33]. This in turn limited their ability to access services such as in-person language classes and healthcare facilities offered outside of their homes [33].

#### 3.3.2. Immigration Policies

A common barrier identified by both refugees and actors working in the field of migrant health was immigration policies, specifically policies that discouraged or delayed care from taking place. Refugees and resettlement agencies described the process of applying for provincial health insurance as lengthy and difficult, which resulted in long periods of refugees being uninsured and unable to receive medical care [22,34]. They also reported having trouble finding clinics and providers who accept refugee medical assistance and reported the complexity of insurance as a barrier present at each point of access in the healthcare system [35]. Healthcare providers expressed significant challenges navigating the excessively complicated and constantly evolving policies that were hard to understand and keep track of [23,24,36]. The lack of institutional support and timely dissemination of information put the onus on individual providers to learn the workings of the complex system to ensure that their patients received appropriate medical care [36,37,38].

Healthcare providers described the healthcare system as being “reactionary” and ill-prepared to provide adequate care for the refugee population [38] (p. 5). Defragmented services resulted in poor information flow, which impeded the providers’ ability to ensure timely care and a continuity of care [37,39,40]. The underuse of available interpreter services was another significant barrier expressed by refugees, providers, and resettlement agencies [31]. Without the means of communication, refugees were sent home without being seen or left the pharmacy without understanding their prescriptions [32,36,41]. Family or community members more proficient in the language were seen to aid during instances without interpreters; however, this tended to compromise the quality of care due to faulty translations or issues disclosing private health information [25,38,42].

#### 3.3.3. Structural Discrimination

Finally, our third most common health barrier identified by refugees was structural discrimination. Our findings show that prejudice and discrimination by the public and among care providers serves as a major barrier. Healthcare workers expressed that the negative images of refugees in the media instigated a general disposition of the public to be rude towards refugees, and that the negative attitudes also extended to the care providers [37]. Refugees reported instances of not being able to make appointments because the receptionist would hang up on the phone from not understanding them [31]. They also reported not feeling respected or valued by the healthcare system, as observed during instances where refugees experienced significant stress related to not being provided with interpreters or explanations about decisions being made about their care [25]. There were reports of healthcare workers becoming angry or making blatantly racist remarks toward refugees [25]. Discrimination was often seen in the form of a dismissal of patients because of limited communication, considering their concerns as trivial, or a dismissal of mental health concerns as resettlement stress [34,43]. In fact, refugee resettlement workers described the whole mental health system as being unavailable to non-English speaking individuals [34].

## 4. Discussion

The purpose of this review was to bridge the gap in the current literature by identifying health barriers faced by documented temporary foreign workers and refugees in high-income OECD countries through a focused lens on the structural origins of poor health outcomes, and to gather information to help guide the planning of a proposed newcomer health clinic in Southern New Brunswick, Canada. We discuss four common emerging themes identified for both documented temporary foreign workers and refugees, while highlighting results that confirm previous findings as well as novel ones deserving attention. Further, we discuss the implications of these findings on their use in healthcare practice and plausible future directions of research.

### 4.1. Housing Conditions

The link between suboptimal housing and poor health outcomes, including stress, depression, and asthma exacerbations is well established in the scientific literature [44]. Our results showed that documented temporary foreign workers and refugees experienced poor housing conditions related to overcrowding and poor hygiene, similar to prior reports [12]. Specifically, new findings from our review demonstrated that previously unstudied populations including documented temporary foreign workers in nonagricultural sectors, as well as refugees with disabilities, also identified poor housing as a key determinant of health. Moreover, this review highlighted notable differences within migrant groups that are important to distinguish in order to better develop interventions including a lack of privacy from employers among caregivers versus a lack of bedding and hygiene among agricultural workers.

### 4.2. Immigration Policies

A growing literature points to the impact immigration policies have on health outcomes [4,6]. Historically, created to define a national belonging and ethnicity, many immigration policies have continued to restrict the rights of migrants [4]. These include discouraging individuals from participating in paid work and denying them protection against unemployment or injuries [45]. Findings from our review demonstrated the direct effects of policy on the health of migrant communities.

In concert with previous studies, immigration policies that required a clean medical record were seen as preventing those desiring permanent residency from seeking care. Our review further explored the detailed implications of such policies, including significant delays in the process of applying for health insurance resulting in long lapses in time where refugees were ineligible for medical care. These results have significant implications and calls for a closer look at the absence of appropriate oversight by institutions that result in migrant workers being vulnerable to abuse and exploitative work conditions. Healthcare workers can also be empowered to advocate for this population in knowing this information and can further advocate for better healthcare access and may inform future policy change considerations.

Our review found a “disconnect” between service agencies and the absence of well-established referral pathways as a major deterrent for the care provision for refugees. In addition, the lack of timely dissemination of information regarding policy changes to care providers was seen as a major constraint in providing timely care. These findings go beyond previous reviews that identified barriers limited to factors within the healthcare system. Our review highlighted how immigration policies outside the healthcare system, society, and institutions in general had a direct impact on creating barriers. This finding corroborates the findings from Bailey et al. [6] (p. 1458) that states the danger of “view[ing] these problems solely as a matter of institutional and interpersonal discrimination within health-care settings” as well as the need to “understand the broad context within which health-care systems operate” (p. 1458) such as intersectoral work guided by transdisciplinary frameworks. Policymakers, shareholders, advocates, and healthcare providers can benefit from knowing this disparity and work to improve intersectoral communication. Our findings outline how creating better pathways to access care can improve health outcomes.

A noteworthy and beneficial finding of this review includes the underuse of available language interpretation services by healthcare providers. Further investigations on the nature of the low uptake in routine clinical care would benefit hospitals and policymakers in ensuring an effective implementation of this important service. In addition, the knowledge of the lack of language services available to patients is important for healthcare providers to be aware of in order to better advocate for services.

### 4.3. Structural Discrimination

Scholars describe a process of “othering” in which minorities are ascribed a lower status on the social hierarchy due to perceived racial and ethnic differentials [4] (p. 2101). People lower on the social hierarchy not only have limited access to social goods but also experience a physiological stress response that increases the risk of chronic diseases [45]. Several papers in our review highlighted the way migrant workers and refugees are regarded and treated as subcitizens in their host countries. This was evident through the marginal living of migrants and a general lack of regard for their health and safety echoed in many of the studies. Acknowledging the role of cultural exclusion and racism would be critical in taking the next step to improving migrant health, as a growing body of literature point to a significant association between racism and poor mental and physical health [46]. Future research may elaborate on the types of microaggression within the healthcare system and how it affects the quality of care migrants receive.

### 4.4. Exploitative Labour Practices

We identified a broader variety of worker occupations, including caregiver, food industry, construction, and hospitality workers. Evidence suggests that job insecurity, occupational hazards, and having no control over high workloads lead to a range of physical and mental afflictions [45]. These findings have implications for the population’s health, as these factors influence health outcomes. Awareness of the challenges faced by temporary foreign workers would allow healthcare providers to serve the population better and have a greater understanding of their needs. In our review, many workers across the board endured hazardous and unsafe working conditions. Large meat-processing factory workers were the only unionized group who reported better experiences because of the advocacy provided by the union. Our findings revealed that little is known about the presence of unions in other parts or industries in Canada. It would be interesting to conduct further research into how and why certain industries were able to establish a union, as well as further characterize how it benefits the health outcomes of documented temporary foreign workers. The present findings also indicate that the current knowledge on the health of documented temporary foreign workers is limited to those surrounding their labor practices and work experiences, with a dearth of studies exploring their use of healthcare services or their experiences in the healthcare settings. Whether that is due to a lack of use or research is an issue for future work to explore.

### 4.5. Limitations

Our review includes several limitations. First, although our study provided a comprehensive review of the experiences of documented temporary foreign workers and refugees in the last decade, our findings should be interpreted with the awareness that it combined studies from multiple high-income OECD countries, with variable policies, societal norms, and healthcare systems. Second, the study focused on peer-reviewed articles written in English, which might have excluded data from studies written in non-English languages. Third, the inclusion of both qualitative and quantitative research designs, although done to increase the scope of reviewed literature, may limit the interpretation of the findings. Fourth, due to the vast nature of the challenges experienced by immigrants, future research should consider conducting a systematic or meta-analysis review, which may provide further nuances and contextualize these findings. Additional future research topics may also include identifying proposed solutions to such barriers and expectant outcomes.

## 5. Conclusions

This review aimed to identify the health barriers faced by documented temporary foreign workers and refugees residing in OECD countries, with a particular focus on the structural origins of differential health outcomes. This was done as the existing literature highlighted the lack of recognition by the public and the majority of the research community in identifying structural origins as a cause of health disparities. In addition to corroborating prior reports, the present review identified structural racism as a direct contributor to poor health outcomes in these populations, as evidenced through studies conducted in the last decade. As there were acknowledged gaps in the literature for subgroups of migrants, we focused on those groups, namely, documented temporary foreign works and refugees. This review will contribute to the growing literature indicating the need to address systemic barriers faced by documented temporary foreign workers and refugees when accessing healthcare services. These findings may benefit healthcare providers, policymakers, and relevant stakeholders interested in developing and improving efforts to eliminate health inequity. In addition, the findings from this review will assist the authors in advocating to relevant authorities about reducing systemic barriers, as they create the proposed newcomer health clinic. Our findings can also inform the newcomer health clinic policies and procedures to make healthcare more accessible to temporary foreign workers and refugees.

## Figures and Tables

**Figure 1 healthcare-11-01295-f001:**
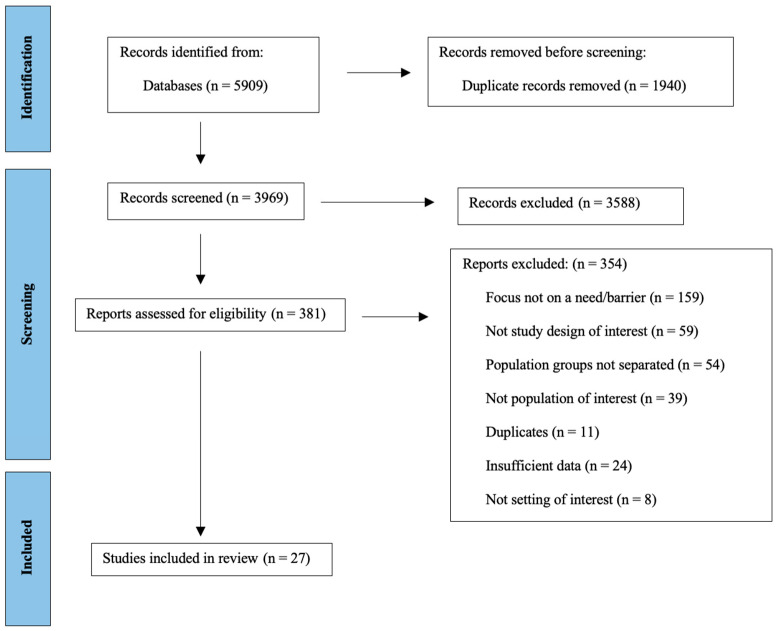
PRISMA flowchart of paper inclusion.

**Table 1 healthcare-11-01295-t001:** PICoS Framework Identifying the Main Focus of Study.

Population	Phenomenon of Interest	Context	Study Type
Documented temporary foreign workers, refugees	Health barriers	OECD high-income countries	Peer-reviewed, primary research

**Table 2 healthcare-11-01295-t002:** Inclusion and Exclusion Criteria for Selection of Papers Included.

Inclusion Criteria	Exclusion Criteria
OECD high-income country	OECD middle- or low-income country/non-OECD country
Documented temporary foreign worker, or refugee population	General or nonspecific migrant population (i.e., undocumented immigrants, asylum seekers, illegal migrants)
Focus of paper is health barriers, challenges, or needs	Paper focused on policy review, changes, or adaptations
Perspective of key informant or population in question	Papers focused on describing disease prevalence
Population of interest is the main focus of the paper	Papers focused on a topic other than healthcare needs, access, barriers, or challenges
Published in English	Papers published in a language other than English
Published between January 2011 and July 2021	Papers published before 2011
Primary research	Reviews, editorials, commentaries, letters, perspectives, news articles, presentations, conference abstracts, symposium summaries, or another type of nonprimary research

**Table 3 healthcare-11-01295-t003:** Selected Excerpts Demonstrating Themes Identified in the Literature.

Documented Temporary Foreign Worker Excerpts	
Theme	Example
Housing conditions	“My house, the family that I live with, it’s stuffed with cameras…it is not like you do something bad to the kids but you have to all the time think about what you are doing—maybe they are not watching me but I think maybe they were; like my self-esteem goes very down and you feel, okay, no I’m not gonna do this thing because may be someone’s watching me” [17] (p. 5). Jose pays 800 dollars to live here. He suggests to the interviewer that a report be written, “a report of what the living conditions are here”. “The boss should out of obligation put a bed, a little one, for each worker. He doesn’t have [provide] anything. So I’m expected to throw myself on the ground” [18] (p. 214).
Immigration policies	The last thing they have in mind is their health issue. And when a certain health issue happen, they try to cover it, right? Because part of having a successful immigration application is having a positive medical exam. There’s a couple that we worked with recently, whose husband contracted a liver issue, and um, and they hid it from, you know (the doctors)? But when they were forced to take the medical exam from the doctor provided by the immigration, they failed. And that’s the basis of their denial … So the whole family is now facing deportation [19] (p. 17).
Structural discrimination	In Beaumont [fictional name], when we were in Beaumont, we couldn’t do [nothing]. If a black guy comes in the community or in the city or in the town of Beaumont—haven’t done anything—the cops would be coming straight to us. It was only 4 black guys […] and everyone, oh those are the Jamaicans [18] (p. 212).
Exploitative labour practices	I got into an accident, because they are supposed to change you every forty-five minutes or every hour to prevent any injury in your body. They are supposed to rotate all the personnel. But they left me there for four hours. So I started having a lot of pain from my neck to my back, and then my shoulder. And I was there for four hours so nobody was changing me … So I deal with them and say, you know what, I have to take some days off and [the manager of the company] told me, no, you can’t because we need you [19,20] (p. 437). I make thirty-two kidney stones, they took me to the hospital and I get surgery”. When he explained the situation to his boss and asked, without success, to work fewer hours in order to rest, “[h]e took my papers, threw them to the floor and said to me ‘your health is not my business’ …” [20] (p. 437).
**Refugee Excerpts**	
**Theme**	**Examples**
Housing conditions	We did not feel safe and secure here at all. When we first arrived we had shots fired on our home. We had to keep the windows and doors locked constantly. We did not dare to go out. Even when we needed to go out to get food, the kids would be too afraid to leave the house [21] (p. 5).
Immigration policies	‘‘I don’t know what is the reason that my daughter doesn’t have Medicaid at the current time, and since two months she is without Medicaid, and she is a child, and why the Medicaid stopped. We filled application for Medicaid two times. First time they asked us to fill it again, and second time before one week we sent another application, and there is no replay or interview appointment or any thing from Medicaid office yet” [22] (p. 1531). “And of course this makes it very difficult in everyday life, so if a doctor first has to think about what the Government agencies will approve and what they will not approve, then, yes, it can be difficult for the medical side of the process” [23] (p. 3). There’s a couple of key disconnects in the [government] system [that need to be] bridged in order to actually do systems level planning for the intake of refugees […] two key disconnects […] are the fact that refugee intake is a federal issue, [while] provision of health and social services is a provincial issue and they’re just not linked [24] (p. 152).
Structural discrimination	The doctor was an old man and when he see our ID [identification card], he said, “oh, from Sudan. Oh, Sudanese, they are the worst people” … I felt really bad. I’m a human like them. I have everything; the only thing that I cannot understand is language [25] (p. 5).

**Table 4 healthcare-11-01295-t004:** Structural Barriers Identified in the Literature for Each Studied Population.

Challenge	Refugees	Temporary Foreign Workers
Housing conditions	X	X
Immigration policies	X	X
Structural discrimination	X	X
Exploitative labour practices		X

## Data Availability

The data that support the findings of this study are available from the first author (B.Y.), upon reasonable request.

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
