# Peer review of "Structural Origins of Poor Health Outcomes in Documented Temporary Foreign Workers and Refugees in High-Income Countries: A Review"

_healthcare, 2023, doi:10.3390/healthcare11091295_

Round 1
Reviewer 1 Report
Thank you for the well-written manuscript. I generally find the justification through the background, the method, results and discussions easy to follow. I have the following comments:
The review included both quantitative and qualitative studies. I think this also needs to be justified, and I can imagine this was necessary to increase the number of reviewed articles. However, the inclusion of both research designs also limits the interpretation of the findings and should be as such listed in the limitation section.
Secondly, I will suggest that the authors do a bit of addition to the conclusions to reflect the different vital findings that set out the review
Reviewer 2 Report
As the authors noted, this is an important topic that continues to be under-addressed. I appreciate their use of the Cochrane model, and especially the deductive thematic analysis approach. The methodology provided seems rigorous and congruent to this approach. My main concern is with the discussion section, where I felt is mostly a recapitulation/repeat of the results, rather than what I would have expected as synthesis of the finding against the proposed problem from the introduction - what to do differently to overcome structural barriers in the creation of the Newcomer health facility. A table of the structural findings and known (or identified gap areas) remediation actions / research areas would be very valuable. A structural model showing the expected outcomes from the proposed changes would also be useful.
More generally:
1) There was some odd pagination going on with this paper, for example page 13 of the PDF shows as page 2.
2) LIne 61 might read "...development OF a proposed Newcomber...". Overall the use of English was well done.
3) Please do not split tables (for example Table 1, Table 2) as it makes it hard to read. Would help if they appeared in referred order as well.
4) Some of the language used is not sufficiently defined. For example, Line 93 "a sufficient number" or line 108 "mentioned it only briefly". Can these references be more precise? Please scan manuscript for other word usage that is similarly oblique.
5) LIne 124 - it is unusual to refer to the reviewers by their initials. Maybe just reviewer 1 and 2?
6) You refer to PRISMA in Figure 1 but not in the body of the text. According to PRISMA, they have a preferred way to cite/refer to their approach. Can this be incorporated into the manuscript?
7) Table 3 is quite long an disrupts the flow of reading the paper. Can this be included as an appendix at the end?
8) Table 4 lists the themes, and then starting on line 196, the paper goes through theme, though in a different order. Can they be in the same order.
9) On line 397 I am unsure what "potentially significant" means given that significant is also a statistical term of art. Perhaps it is a finding worth further exploration?
Reviewer 3 Report
The paper in its current version needs some clarification before being published.
a) Justify why they did not use databases such as SCOPUS or WOS, where they could have found a diversity of more multidisciplinary papers.
b) Justify why the period was from January 1, 2011, to July 30, 2021. What would be the methodological reason
c) What gaps does the article cover, since when carrying out a synthetic review of the literature, the evidence presented in the document comes from research carried out by other authors
d) What are the implications of your study?
e) Suggestion to consider: the word “fast” in the title does not seem ideal for the paper
Round 2
Reviewer 3 Report
The paper now reads better with the corrections made by the authors. Thanks
Author Response
Thank you for your valuable comment, we greatly appreciated the guidance provided.